# NET-02 trial protocol: a multicentre, randomised, parallel group, open-label, phase II, single-stage selection trial of liposomal irinotecan (nal-IRI) and 5-fluorouracil (5-FU)/folinic acid or docetaxel as second-line therapy in patients with progressive poorly differentiated extrapulmonary neuroendocrine carcinoma (NEC)

Zoe Craig [ORCID],[1] Jayne Swain,[1] Emma Batman,[1] Jonathan Wadsley,[2] Nicholas Reed,[3] Olusola Faluyi,[4] Judith Cave,[5] Rohini Sharma,[6] Ian Chau,[7] Lucy Wall,[8] Angela Lamarca,[9,10] R Hubner,[9,10] Wasat Mansoor,[9] Debashis Sarker,[11] Tim Meyer,[12] David A Cairns,[1] Helen Howard,[1] Juan W Valle,[9,10] Mairéad G McNamara[9,10]

For numbered affiliations see end of article.

**Correspondence to**
Dr Mairéad G McNamara;
Mairead.McNamara@christie.nhs.uk

## ABSTRACT

**Introduction** Poorly differentiated (PD), extrapulmonary (EP), neuroendocrine carcinomas (NECs) are rare but aggressive neuroendocrine neoplasms. First-line treatment for advanced disease is an etoposide and platinum-based chemotherapy combination. There is no established second-line treatment for patients with PD-EP-NEC, and this is an area of unmet need.

**Methods and analysis** NET-02 is a UK, multicentre, randomised (1:1), parallel group, open-label, phase II, single-stage selection trial of liposomal irinotecan (nal-IRI)/5-fluorouracil (5-FU)/folinic acid or docetaxel as second-line therapy in patients with progressive PD-EP-NEC. One hundred and two eligible participants will be randomised to receive either nal-IRI/5-FU/folinic acid or docetaxel. The primary objective is to determine the 6-month progression-free survival (PFS) rate. The secondary objectives of this study are to determine PFS, overall survival, objective response rate, toxicity, quality of life and whether neuron-specific enolase is predictive of treatment response. If either treatment is found to have a 6-month PFS rate of at least 25%, that treatment will be considered for a phase III trial. If both treatments meet this target, prespecified selection criteria will be applied to establish which treatment to take forward.

**Ethics and dissemination** This study has ethical approval from the Greater Manchester Central Research Ethics Committee (reference no. 18/NW/0031) and clinical trial authorisation from the Medicine and Healthcare Products Regulatory Agency. Results will be published in peer-reviewed journals and uploaded to the European Union Clinical Trials Register.

**Trial registration numbers** ISRCTN10996604, NCT03837977, EudraCT Number: 2017-002453-11

> **Strengths and limitations of this study**
>
> ► The trial is designed to ensure, with a high probability, that the most efficacious treatment is selected to be taken forward to a phase III trial.
> ► Prospectively defined decision criteria in this trial will enable earlier planning of a phase III trial if these targets are reached.
> ► The trial is not powered to directly compare the two treatment arms in this study.

## INTRODUCTION
### Neuroendocrine carcinomas
Neuroendocrine carcinomas (NECs) are a rare, high-grade, poorly differentiated (PD) form of neuroendocrine neoplasms (NENs).[1] The annual incidence of PD extrapulmonary (EP) NEC is approximately one diagnosis per 100 000 persons.[2 3] These tumours are characterised by aggressive histological features; high Ki-67 index (>20% by definition, but usually higher (>75%)),[4] extensive necrosis

and nuclear atypia, and are classified as NEC grade 3 according to WHO 2010 classification.[5]

First-line treatment for PD-EP-NECs has remained largely unchanged since a study in the early 1990s reported antitumour activity and high tumour response rates (RRs) produced by an etoposide–platinum combination.[6] Nevertheless, disease progression invariably occurs in patients during or following completion of first-line therapy, and a standard second-line treatment is yet to be determined.

## Current second-line treatment options for patients with a NEC diagnosis

For patients with advanced PD-EP-NEC, combination regimens such as irinotecan, 5-fluorouracil (5-FU) and folinic acid are a second-line treatment option currently used, without robust trial evidence.[7] This combination has been recommended for patients with a NEC diagnosis with a Ki-67 ≥55%, whereas some literature recommends temozolomide-based combinations for those with a Ki-67 <55%.[8 9] In devising treatment strategies for PD-EP-NEC, many refer to the extensive literature on high-grade NEC of the lung, for which docetaxel is a second-line therapy option.[9]

Several small retrospective studies have published results for the outcomes of second-line chemotherapy after failure of the etoposide–platinum combination in patients with grade 3 NECs.[7–13] The NORDIC-NEC study reported predictive and prognostic factors for treatment and survival in 305 patients with advanced gastrointestinal NEC.[9] Second-line chemotherapy was administered to 100 patients; of these, 35 received temozolomide-based chemotherapy and 20 received docetaxel-based chemotherapy. Of 84 evaluable patients, the RR was 18%. Those whose tumours had a Ki-67 <55% had a lower RR, but better survival than patients whose tumours had a Ki-67 ≥55%. The median overall survival (OS) for patients treated with first-line platinum-based chemotherapy in the advanced setting is 11–16.4 months.[9 14] In a systematic review and meta-analysis of second-line treatment in 595 patients with advanced PD-EP-NEC, the median RR was 18%, the median progression-free survival (PFS) was 2.5 months (range 1.2–6.0) and the median OS was 7.6 months (range 3.2–22).[15]

## Liposomal irinotecan

Irinotecan, a topoisomerase I inhibitor, works to arrest uncontrolled cell growth by preventing the unwinding of DNA, therefore, preventing cell replication and tumour growth.[16]

Liposomal irinotecan (nal-IRI) (ONIVYDE, Servier) is irinotecan encapsulated in a liposome drug delivery system. This stable liposome formulation of irinotecan has several attributes that may provide an improved therapeutic index; controlled and sustained release, high intravascular drug retention and enhanced permeability.[16 17] The improved pharmacokinetics and biodistribution of nal-IRI in comparison to irinotecan may have clinical benefit in patients with NEC.

Pharmacokinetic studies have demonstrated that once irinotecan is released from the liposomes, the conversion to the active metabolite, SN-38, is similar to that of unencapsulated irinotecan.[16 18] Thus, nal-IRI and unencapsulated irinotecan have demonstrated similar adverse reactions (ARs) in patients, the most common of which include gastrointestinal events and myelosuppression.[16 18]

## Rationale for the use of nal-IRI in combination with 5-FU and folinic acid

Preclinical evidence supports the hypothesis that nal-IRI modifies the tumour microenvironment in a manner that should make tumours more susceptible to 5-FU/folinic acid, through decreasing tumour hypoxia and increasing small molecule perfusion.[19 20]

Given the relative absence of overlapping toxic effects among nal-IRI, 5-FU and folinic acid, a regimen combining these agents was studied in a phase I, dose-escalation trial of solid tumours.[21] Among the 15 efficacy-evaluable participants, the overall disease control rate was 73.3%. Among the six participants who received the nal-IRI maximum tolerated dose of $80\,mg/m^2$, the objective RR (ORR) and disease control rate were 16.7% and 83.3%, respectively.

In the NAPOLI-1 phase III trial of nal-IRI, with or without 5-FU and folinic acid, versus 5-FU and folinic acid alone, in the treatment of patients with metastatic pancreatic ductal adenocarcinoma after receiving gemcitabine-based therapy, an increase in OS for those treated with a combination of nal-IRI and 5-FU/folinic acid was reported compared with those treated with 5-FU and folinic acid alone (HR for survival 0.67, 95% CI 0.49 to 0.92).[22]

## Rationale for the use of docetaxel

The National Comprehensive Cancer Network Clinical Practice Guidelines in Oncology, for the treatment of small cell and non-small cell lung cancer, include docetaxel as a second-line treatment option in patients who have progressed on primary etoposide–platinum combination therapy.[23 24] Based on observed RRs, survival, quality of life (QoL) and toxicities, the optimal dose of docetaxel in pretreated patients with non-small cell lung cancer is $75\,mg/m^2$ every 3 weeks.[25]

## Study rationale and AIM

Treatment of patients with advanced PD-EP-NEC, to date, has been analogous to that of high-grade NEC (small cell or non-small cell cancer) of the lung.[6] The standard arm of NET-02 is that used in high-grade lung NEC, of which docetaxel is a second-line therapy option,[23] and combination regimens such as irinotecan/5-FU are a second-line therapy option currently used, without trial evidence, for this subset of patients.[7] Prospective collaborative trials, with translational endpoints, are warranted and may inform future biomarker-driven studies.

Therefore, the overall aim of this trial is to assess the efficacy of nal-IRI/5-FU/folinic acid or docetaxel, separately, as second-line therapy in patients with progressive PD-EP-NEC, with selection criteria applied to establish which treatment to take forward into a phase III trial.

## METHODS AND ANALYSIS

### Trial objectives

The primary objective of the trial is to determine the 6-month PFS rate, defined as a binary outcome (progression free or not) within the time frame of treatment start date until 6 months postrandomisation.

The secondary objectives of the trial are to determine:

▶ PFS (defined as the time from randomisation to progression or death from any cause).

▶ OS (defined as the time from randomisation to death from any cause).

▶ ORR at 6 months postrandomisation (defined using the Response Evaluation Criteria in Solid Tumours V.1.1 measurements).[26]

▶ Toxicity as per Common Terminology Criteria for Adverse Events (CTCAE) V.5.0.

▶ QoL (defined using European Organisation for Research and Treatment of Cancer (EORTC) QoL validated questionnaires (QLQ) C30 (EORTC QLQ-C30)[27] and GINET21 (EORTC QLQ-GINET21)).[27 28]

▶ Association between neuron-specific enolase concentration and treatment response.

Additional exploratory objectives, analysing participant samples, will include:

▶ Quantification of circulating tumour cells (CTCs) and circulating tumour DNA at baseline, 6 weeks and on progression, to identify any correlation with disease-related outcomes.

▶ Molecular profiling of CTCs, circulating tumour DNA and tumour tissue (further immunohistochemistry on tumour tissue may also be required) to identify any correlation with disease-related outcomes.

▶ Generation of mouse models of PD-EP-NEC.

### Trial design

The NET-02 trial is a UK, multicentre, randomised (1:1), parallel group, open-label, phase II, single-stage selection trial of nal-IRI/5-FU/folinic acid or docetaxel as second-line therapy in patients with progressive PD-EP-NEC.

The design is an adaptation of a one-stage trial design proposed by Simon *et al*, where the A'Hern design is first implemented to assess efficacy of each treatment separately, to ensure a prespecified minimum level of activity prior to selection.[29] Should both treatments be sufficiently efficacious, prespecified selection criteria are then applied to establish which treatment to take forward into a phase III trial. The intention of the trial is to show that the regimens are sufficiently active in this patient population, but not to show that one regimen is significantly superior to the other.

The A'Hern method is advantageous over other single-stage designs, since it uses the exact binomial distribution, as opposed to a normal approximation to the binomial distribution which can lead to substantial error in small trials.[30] Additionally, prospectively defined decision criteria, specified below, are applied, which if reached, could enable earlier planning for a phase III follow-on trial.

Participants will be randomised to receive either nal-IRI/5-FU/folinic acid, administered every 14 days or docetaxel, administered every 21 days. Trial treatment will continue until progressive disease, intolerable toxicity, delay of treatment for more than 28 days, development of any condition or occurrence of any event, which, in the opinion of the local investigator, justifies discontinuation of treatment, participant request or until 6 months after the last participant is randomised, whichever occurs first. Figure 1 displays the full trial schema.

### Trial population and sample size

The NET-02 trial will recruit patients diagnosed with PD-EP-NEC (Ki-67 >20% and grade 3, confirmed by histology). Patients will be eligible for the trial if they meet all of the inclusion criteria and do not satisfy any of the exclusion criteria listed in table 1.

One hundred and two eligible participants will be randomised to receive either nal-IRI/5-FU/folinic acid or docetaxel. Allowing for a 5% drop-out rate, this will provide 80% power for demonstrating that the one-sided 95% CI for the 6-month PFS rate excludes 15%, if the true rate is at least 30%, where 30% is the required level of efficacy and a rate of 15% or less would give grounds for rejection, that is, the relevant treatment would be considered not to have reached an acceptable level of efficacy to warrant further evaluation. The proportions of 15% and 30% were chosen in line with existing literature; of those who reported the proportion progression free at 6 months, the lowest was approximately 15%[11] and the highest approximately 25%.[7] Therefore, for either trial treatment to be taken forward for further research, they should provide estimates that are at least as good as the lower value and aim to improve on the higher value.

A treatment arm may be considered for further evaluation using the treatment selection process described below, if at least 12 out of 48 evaluable participants are progression free at 6 months (equating to a success rate of 25%, with a lower one-sided 95% confidence limit of 15.1%).

### Treatment selection

If both treatments successfully exceed the predefined criteria, having lower one-sided 95% confidence limits greater than 15%, Simon *et al*'s design proposes that the treatment with the higher PFS rate at 6 months should be selected, regardless how small its advantage over the other treatment appears.[29] Nevertheless, to ensure that the more efficacious treatment is selected with a high probability, if the difference in the 6-month PFS rates is less than 5%, additional selection criteria, including toxicity rates and QoL score, will be considered. If only one of the

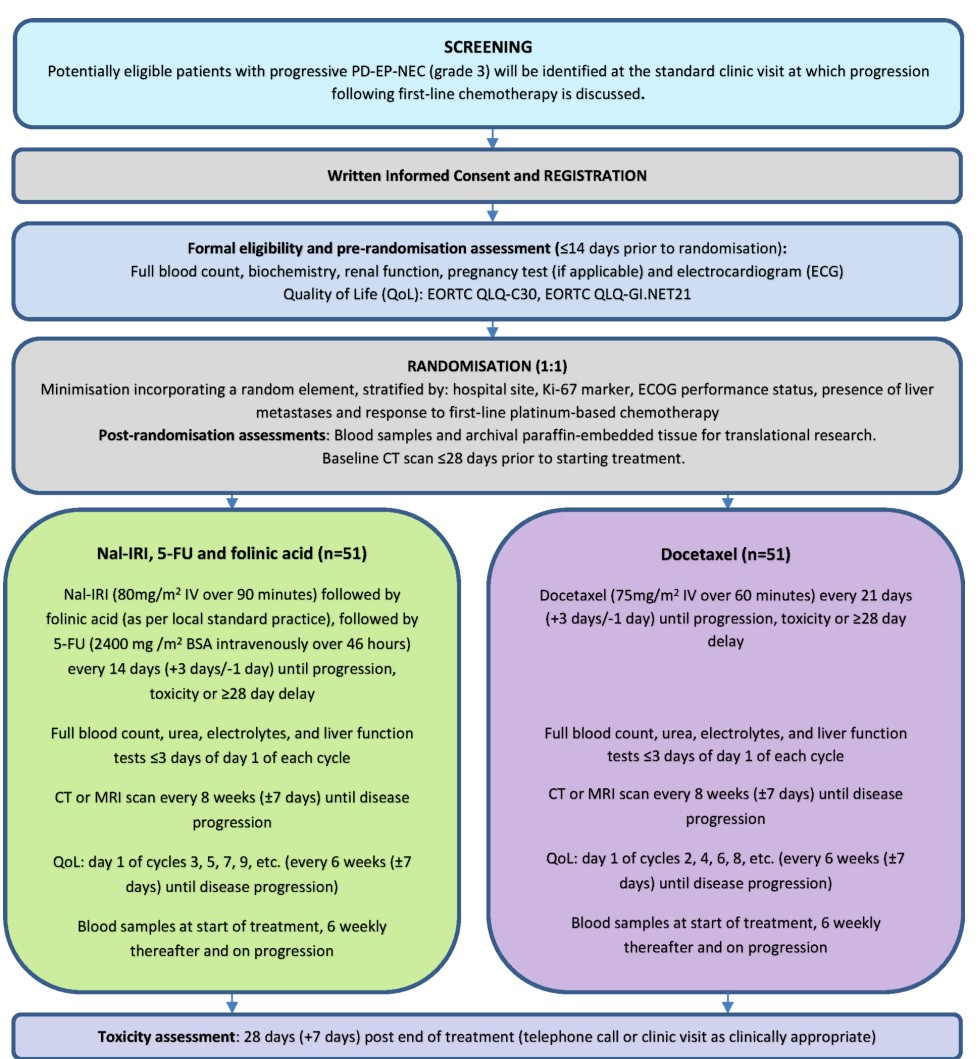

**Figure 1** Trial schema. ECOG, Eastern Co-operative Oncology Group; EP, extrapulmonary; EORTC, European Organisation for Research and Treatment of Cancer; 5-FU, 5-fluorouracil; NEC, neuroendocrine carcinoma; nal-IRI, liposomal irinotecan; PD, poorly differentiated.

treatments successfully exceeds the predefined criteria, this treatment will be selected for further investigation.

### Recruitment, registration and randomisation

Participants will be recruited from 16 UK sites (see online supplementary material) over a 37-month period. Potential participants will be approached, regarding trial participation, during the standard clinic visit at which their progression following first-line chemotherapy is discussed and will be provided with a verbal and written explanation of the trial. Patients, who provide written informed consent, to the site principal investigator or delegate, will be registered onto the trial. Consent to the use of blood samples for future projects and mouse model generation (The Christie National Health Service (NHS) Foundation Trust participants only) is optional.

Recruitment of participants to the NET-02 trial requires trial-specific investigations to confirm eligibility. Consequently, recruitment is a two-step process involving registration and randomisation.

Initial registration will involve all patients who have provided written informed consent. Patients will undergo investigations to confirm eligibility including a full blood count, biochemistry and renal function assessment, an ECG and a pregnancy test (if applicable) to confirm that they satisfy the eligibility criteria specified in table 1.

Once all other screening investigations are successfully completed and prior to meeting with the clinician and randomisation, two baseline QoL questionnaires (EORTC QLQ-C30[27] and EORTC QLQ-GINET21[28]) will be completed.

Patients identified as eligible, following the eligibility assessments, will be randomised. If more than 14 days have elapsed since the initial eligibility blood tests, these must be repeated prior to randomisation, to ensure that the patient remains eligible. Registration and randomisation will be performed centrally using either the Leeds Clinical Trials Research Unit (CTRU) automated telephone or web-based system.

**Table 1** NET-02 inclusion and exclusion criteria

| Inclusion criteria | Exclusion criteria |
|---|---|
| 1. Age≥18 years and life expectancy >3 months. <br> 2. Diagnosed with poorly differentiated (as defined by WHO in 2010, Ki-67 >20%) EP-NEC (grade 3, confirmed by histology). Carcinoma of unknown primary is allowed if lung primary has been excluded following review by the multidisciplinary team. <br> 3. Prior treatment with first-line platinum-based chemotherapy for NEC in the advanced setting and ≥28 days from day 1 of the previous treatment cycle. <br> 4. Documented radiological evidence of disease progression OR discontinuation of first-line platinum-based chemotherapy due to intolerance. <br> 5. Measurable disease according to RECIST V.1.1. <br> 6. Eastern Co-operative Oncology Group performance status ≤2. <br> 7. Adequate renal function with serum creatinine ≤1.5 times upper limit of normal (ULN) and creatinine clearance ≥30 mL/min according to Cockroft-Gault or Wright formula. If the calculated creatinine clearance is <30 mL/min, glomerular filtration rate (GFR) may be assessed using either Cr51-EDTA or 99mTc-DTPA clearance method to confirm if GFR is ≥30 mL/min). <br> 8. Adequate haematological function: Hb ≥90 g/L, WCC≥3.0×10⁹/L, ANC≥1.5×10⁹/L, platelet count ≥100×10⁹/L. <br> 9. Adequate liver function: serum total bilirubin ≤1.5 x ULN (biliary drainage is allowed for biliary obstruction) and ALT and/or AST≤2.5 x ULN in the absence of liver metastases, or ≤5 x ULN in the presence of liver metastases. <br> 10. A negative pregnancy test is required at registration in women of childbearing potential. <br> 11. Men and women of reproductive potential must agree to use a highly effective form of contraception during the study and for 6 months following the last dose of trial treatment. In addition, male participants should use a condom during study participation and for 6 months following the last dose of trial treatment. <br> 12. Patients must be able to provide written informed consent. <br> 13. Patients must be able and willing to comply with the terms of the protocol. | 1. Known or suspected allergy or hypersensitivity reaction to any of the components of study treatment or their excipients. <br> 2. Use (including self-medication) within 1 week of randomisation and for the duration of the study of any of the following: St. John's wort, grapefruit, Seville oranges, medicines known to inhibit UGT1A1 (eg, atazanavir, gemfibrozil, indinavir) and medicines known to inhibit or induce either CYP3A4 or CYP3A5. <br> 3. Previous treatment (for NEC) with any of the components of combination chemotherapy regimens detailed in this study (nal-IRI, 5-FU, irinotecan, topoisomerase inhibitors or taxane-based therapy). <br> 4. Incomplete recovery from previous therapy in the opinion of the investigator (surgery/adjuvant therapy/radiotherapy/chemotherapy in advanced setting), including ongoing peripheral neuropathy of >Common Terminology Criteria for Adverse Events (CTCAE) V.5.0 grade 2 from previous platinum-based therapy. <br> 5. Concurrent palliative radiotherapy involving target lesions used for this study (<28 days from discontinuation of radiotherapy). Radiotherapy for non-target lesions is allowed if other target lesions are available outside the involved field. <br> 6. Patients must not have a history of other malignant diseases (within the previous 3 years, and there must be no evidence of recurrence), other than: <br> – EP-NEC. <br> – Non-melanoma skin cancer where treatment consisted of resection only or radiotherapy. <br> – Ductal carcinoma in situ where treatment consisted of resection only. <br> – Cervical carcinoma in situ where treatment consisted of resection only. <br> – Superficial bladder carcinoma where treatment consisted of resection only. <br> 7. Documented brain metastases, unless adequately treated (surgery or radiotherapy only), with no evidence of progression and neurologically stable off anticonvulsants and steroids. <br> 8. Clinically significant gastrointestinal disorder (in the opinion of the treating clinician), including hepatic disorders, bleeding, inflammation, obstruction or diarrhoea >CTCAE grade 1 (at time of study entry). <br> 9. Severe arterial thromboembolic events (myocardial infarction, unstable angina pectoris, stroke) less than 6 months before inclusion. <br> 10. New York Heart Association class III or IV congestive heart failure, ventricular arrhythmias or uncontrolled blood pressure. <br> 11. Severe bone marrow failure or bone marrow depression after radiotherapy or treatment with other antineoplastic agents (defined as haematological values of Hb or white blood cells or neutrophils or platelets not meeting inclusion criteria). <br> 12. Known active hepatitis B virus, hepatitis C virus or HIV infection. <br> 13. Active chronic inflammatory bowel disease. <br> 14. Breastfeeding women. <br> 15. Evidence of severe or uncontrolled systemic diseases, which, in the view of the treating clinician, makes it undesirable for the patient to participate in the trial. <br> 16. Evidence of significant clinical disorder or laboratory finding which, in the opinion of the treating clinician, makes it undesirable for the patient to participate in the trial. <br> 17. Medical or psychiatric conditions that impair the ability to give informed consent. <br> 18. Any other serious uncontrolled medical conditions (in the opinion of the treating clinician). |

ALT, alanine aminotransferase; ANC, absolute neutrophil count; AST, aspartate aminotransferase; EDTA, ethylenediaminetetraacetic acid; EP, extrapulmonary; 5-FU, 5-fluorouracil; Hb, haemoglobin; nal-IRI, liposomal irinotecan; NEC, neuroendocrine carcinoma; RECIST, Response Evaluation Criteria in Solid Tumours; UGT, uridine diphosphate-glucuronosyl transferase; WCC, white cell count.

A minimisation programme, which incorporates a random element, will be used for randomisation to ensure treatment groups are well balanced for the following characteristics:
► Hospital site.
► Ki-67 marker (<55%, ≥55%).
► Eastern Co-operative Oncology Group (ECOG) performance status (0/1, 2).
► Presence of liver metastases (yes, no).

► Response to first-line platinum-based chemotherapy (resistant disease (progression ≤6 months from completion of platinum-based therapy), sensitive disease (progression >6 months from completion of platinum-based therapy), platinum intolerant).

Following randomisation, baseline assessments will be conducted. These will include; medical history, demographics, baseline symptoms, physical examination, vital signs, CT scan (or MRI scan, if appropriate) of the thorax-abdomen-pelvis and staging within 28 days of starting trial treatment, one 10 mL blood sample for local measurement of neuron-specific enolase and two 10 mL blood samples for central translational research. Confirmation of availability of archival paraffin-embedded tissue for translational research will also be sought. An additional 10 mL blood sample may be taken for mouse model development for consenting participants from The Christie NHS Foundation Trust.

## Interventions
Nal-IRI (ONIVYDE, Servier), folinic acid and 5-FU will be administered sequentially. The recommended dose and regimen of nal-IRI is $80 \, mg/m^2$ body surface area (BSA) intravenously over 90 min ($\pm$10 min), followed by folinic acid as per local standard practice (recommended dose is 350 mg fixed dose), followed by 5-FU 2400 mg/$m^2$ BSA intravenously over 46 hours. Following cycle 1, subsequent doses will be administered every 14 days (+3 days/−1 day). Where it is not possible to administer nal-IRI due to toxicity, 5-FU/folinic acid can be administered as a monotherapy.

Docetaxel will be administered at a dose of $75 \, mg/m^2$ BSA as an intravenous infusion over 60 min, or as per local standard practice. Following cycle 1, subsequent doses will be administered every 21 days (+3 days/−1 day).

Dosing may be postponed for up to 28 days from when it was due, to allow for (but not limited to) recovery from treatment-related toxicities, infection or following patient request. In the event of a delay due to toxicity, a dose modification (see online supplementary material) may be required at subsequent cycles following a dose delay. If a patient's dose is reduced due to toxicity, it will remain reduced for the duration of treatment. Patients who have already received two dose reductions and experience additional toxicities that would require further dose reduction should discontinue study medication. However, in the event that the participant is deriving clinical benefit and the treating clinician would prefer to continue treatment, an additional dose reduction may be permitted at the discretion of the chief investigator or delegate. If the toxicity recovery duration (to ≤grade 2 CTCAE V.5.0 or baseline) is more than 28 days, the participant should discontinue trial treatment. Participants who have prematurely discontinued treatment will continue to attend 8-weekly clinic visits for CT scans and have follow-up data collected, unless the participant withdraws consent for follow-up visits and further data collection.

All concurrent medical conditions and complications of the underlying malignancy will be treated at the discretion of the treating physician according to local standards of medical care. Participants can receive analgesics, antiemetics, antibiotics, antipyretics and blood products as necessary. However, the use of warfarin-type anticoagulant therapies is not permitted.

## Treatment cycle assessments
Participants on the nal-IRI/5-FU/folinic acid treatment arm will have 2-weekly treatment cycles. Participants on the docetaxel treatment arm will have 3-weekly treatment cycles.

Assessments carried out on the first day of each treatment cycle will include; laboratory assessments, clinical evaluation, vital signs, ECOG performance status, physical examination, details of concomitant medication and toxicity assessment (from cycle 2 onwards). Translational research blood samples and QoL questionnaires will be collected at 6-weekly intervals and at disease progression. A CT or MRI scan will be carried out 8 weekly (±7 days) from treatment start until disease progression or until 6 months after the last participant is randomised, whichever occurs first. Disease progression will be defined as the date of the CT or MRI scan that identifies disease progression. In the rare circumstances that disease progression is determined clinically and it is not appropriate to confirm it radiologically, the date of progression will be defined as the date of documented clinical disease progression.

## Safety
Adverse events (AEs) and ARs will be collected on the first day of each treatment cycle from cycle 2 onwards. Serious AEs (SAEs), serious ARs (SARs) and suspected unexpected serious ARs (SUSARs) will be collected from registration. All AEs, ARs and SAEs will be collected until 28 days after the last dose of trial treatment was administered; SARs and SUSARs will be collected until the end of the study.

## Data collection
Data will be collected using paper case report forms and entered into a validated trial database by the CTRU, where data quality will be monitored. Automatic and manual validation of entered data will be conducted. Data items relating to the safety and rights of individual participants will be dealt with as a priority. Data items required for the primary endpoint analysis will be manually checked at the CTRU. Missing data will be chased until it is either received or confirmed as not available at the trial analysis stage.

## Statistical analysis
A full statistical analysis plan will be written before any analysis is undertaken.

The primary analysis population will be defined as those who have received at least one dose of the protocol treatment. Individuals will be analysed according to the treatment that they received rather than that which they

were randomised to receive. The QoL population is defined as any individual who returned at least one QoL questionnaire. Unless otherwise stated, the analysis will be conducted separately for each treatment group as per the primary analysis population.

All analyses will use a 5% significance level. The primary endpoint will be presented with a one-sided CI, while secondary endpoints will be presented with two-sided CIs. No formal interim analyses are scheduled to occur; hence, no statistical testing will take place until final analysis, which will occur once all randomised participants have reached the primary endpoint. Nevertheless, the Data Monitoring and Ethics Committee (DMEC) will receive full reports, at least annually and safety reports at least 6-monthly, to monitor participant safety and trial progress, and they may prematurely terminate the trial if necessary.

Primary endpoint analysis of the proportion of participants progression free at 6 months postrandomisation will be calculated using exact methods. If the one-sided CI for either treatment from this analysis includes 15%, then that treatment will not be considered for a phase III trial. An individual is defined to have achieved the primary endpoint if they do not progress within the time frame of treatment start date until 6 months postrandomisation. If an individual dies or is lost to follow-up, without confirmation of disease progression, within 6 months postrandomisation, they will be considered to have not achieved this endpoint and will be censored at the date of death or date last known to be alive and progression free.

Secondary endpoint analysis will include summary statistics and Kaplan-Meier survival curves for PFS and OS, summaries of the number and cause of deaths, and calculation of the ORR (defined as the proportion of participants achieving at least a partial response within 6 months postrandomisation).

Safety analyses will summarise AEs, ARs, SAEs, SARs, SUSARs and pregnancies. Line listings of SAEs will be generated and will include details on expectedness, causality, relationship to the trial treatment and outcome.

QoL will be summarised using mean scores for each subscale and repeated measures models will be employed to investigate changes in health-related QoL over time for each treatment group, using the QoL population.

In the event that both treatment groups meet the specified threshold for the primary endpoint, and show a similar level of efficacy, toxicity and QoL data will inform which treatment to investigate in further research.

Summary statistics for the concentration of neuron-specific enolase at each time point will be estimated. The baseline concentration of neuron-specific enolase will be analysed to assess whether it is associated with response to treatment at 6 months postrandomisation, via an ordinal logistic regression model, adjusting for the stratification factors (excluding hospital site) and any appropriate interaction variables.

Exploratory analysis of the primary and selected secondary endpoints (PFS, OS and ORR) will be done using logistic or Cox regression, as appropriate. All models will be adjusted for the stratification factors (excluding hospital site). Subgroup analysis of the primary and selected secondary endpoints (as above) will include investigation of gender, age, Ki-67 value and morphology of NEC. All exploratory and subgroup analyses will be considered as hypothesis-generating rather than as confirmatory if significant differences are found. Further exploratory and subgroup analyses beyond that described may be undertaken.

### Trial monitoring
A trial monitoring plan will be developed by the trial management group (TMG) and agreed by the trial steering committee (TSC), based on the trial risk assessment. The TMG, comprising the chief investigator, CTRU team, other key trial staff, a nursing representative and a patient and public involvement (PPI) representative will be assigned responsibility for the clinical setup, ongoing management, promotion of the trial and the interpretation and publishing of the results. The TSC and DMEC will provide independent oversight of the study and will be responsible for monitoring the study conduct. The TSC, comprising a statistician, an oncologist and a PPI representative will provide overall supervision of the trial. The DMEC, composed of two gastroenterologists, an oncologist (all with experience in the treatment of patients with NENs) and a statistician, will review the safety and ethics of the study alongside the trial progress, as overseen by the TSC. The DMEC will review confidential safety reports at least 6-monthly and the DMEC and TSC will meet separately, at least annually, to discuss trial progress.

### Patient and public involvement
PPI representatives are involved in the design and overall direction of the trial through their roles in the TMG and the TSC. As part of the TMG, the PPI representative has been involved in protocol development and the preparation of the patient information and informed consent trial documentation. As part of the TSC, the PPI representative will provide advice regarding trial design and conduct, and will be involved in monitoring trial progress and patient safety.

### Ethics and dissemination
The NET-02 trial opened to recruitment on 16 November 2018. At the time of submission, 12 centres out of 16 are open to recruitment, and 17 participants have been randomised into the trial. The trial is currently adhering to V.3.0 of the protocol (approved 20 September 2018), with all sites opening to this version of the protocol. The trial is sponsored by The Christie NHS Foundation Trust, coordinated by Leeds CTRU and funded by Servier (unrestricted grant).

The trial will be conducted in accordance with Good Clinical Practice. Trial results will be published in peer-reviewed journals and will be reported in line with the

Consolidated Standards of Reporting Trials guidelines.[31] Authorship will be decided according to the International Committee of Medical Journal Editors criteria for authorship.[32] All publications will be reviewed by the sponsor and funder prior to publication. To maintain the scientific integrity of the trial, data will not be released prior to the first publication of the analysis of the primary endpoint, for either trial publication or oral presentation purposes, without the permission of the sponsor and TSC. Research results will also be uploaded to the European Union Clinical Trials Register.

All information collected during the course of the trial will be kept strictly confidential. Information will be held securely at the CTRU. The CTRU will comply with all aspects of the General Data Protection Regulation 2018.[33] The trial staff at the participating sites will be responsible for ensuring that any data or documentation sent to the CTRU is appropriately anonymised. At the end of the trial, data will be securely archived in line with the sponsor's procedures for a minimum of 15 years. Data held by the CTRU will be archived in the sponsor archive facility, and site data and documents will be archived at the sites. Following authorisation from the sponsor, arrangements for confidential destruction will then be made.

**Author affiliations**
[1]Clinical Trials Research Unit, Leeds Institute of Clinical Trials Research, University of Leeds, Leeds, UK
[2]Department of Oncology, Weston Park Hospital, Sheffield, UK
[3]Beatson West of Scotland Cancer Centre, Glasgow, UK
[4]Clatterbridge Cancer Centre NHS Foundation Trust, Bebington, UK
[5]Department of Oncology, University Hospital Southampton NHS Foundation Trust, Southampton, UK
[6]Department of Surgery and Cancer, Imperial College London, London, UK
[7]Gastrointestinal and Lymphoma Unit, Royal Marsden Hospital NHS Trust, London, UK
[8]Department of Oncology, Western General Hospital, Edinburgh, UK
[9]Department of Medical Oncology, The Christie NHS Foundation Trust, Manchester, UK
[10]Division of Cancer Sciences, The University of Manchester, Manchester, UK
[11]Comprehensive Cancer Centre, King's College Hospital, London, UK
[12]Department of Oncology, University College London Cancer Institute, London, UK

**Acknowledgements** Thanks to Alison Backen (Project Manager, The Christie NHS Foundation Trust) for her contribution to initial protocol development.

**Contributors** Conception and design of the NET-02 trial: JS, DAC, HH, JWV and MGM. Development of the protocol and patient information sheet: JS, DAC, HH, DS, OF, TM, JW and MGM. Writing of manuscript: ZC, JS, DAC and MGM. Review of manuscript: EB, JW, NR, OF, JC, RS, IC, LW, AL, RAH, WM, DS, TM and HH. All authors read and approved the final manuscript.

**Funding** This research is investigator initiated and funded by an unrestricted educational grant from Servier (grant reference number 016-34263). This work was also supported by Core Clinical Trials Unit Infrastructure from Cancer Research UK (C7852/A25447). SPONSOR: The Christie NHS Foundation Trust, Wilmslow Road, Manchester, M20 4BX, UK. Sponsor reference: CFTSp116.

**Competing interests** JW reports grants and personal fees from AstraZeneca, grants and personal fees from SanofiGenzyme, personal fees and non-financial support from Celgene, personal fees from Eisai, personal fees and non-financial support from Ipsen, personal fees and non-financial support from Novartis, non-financial support from Imaging Equipment, outside the submitted work. IC reports advisory role for Eli-Lilly, Bristol Meyers Squibb, MSD, Bayer, Roche, Merck-Serono, Five Prime Therapeutics, AstraZeneca, Oncologie International, Pierre Fabre;

research funding from EliLilly, Janssen-Cilag, Sanofi Oncology, Merck-Serono; honorarium from Eli-Lilly. AL received travel and educational support from Ipsen, Pfizer, Bayer, AAA, Sirtex Medical, Novartis, Mylan and Delcath Systems; speaker honoraria from Merck, Pfizer, Ipsen and Incyte; advisory honoraria from EISAI and Nutricia; she is a member of the Knowledge Network and NETConnect Initiatives funded by Ipsen. DS reports personal fees from MSD, personal fees and non-financial support from EISAI, personal fees and non-financial support from Ipsen, personal fees from Bayer, non-financial support from Mina Therapeutics, personal fees from Pfizer, personal fees from Novartis, outside the submitted work. TM reports grants from Bayer, grants from BTG, personal fees from BMS, personal fees from EISAI, personal fees from AstraZeneca, personal fees from Tarveda, personal fees from Ipsen, personal fees from MSD, outside the submitted work. DAC reports grants and non-financial support from Servier, during the conduct of the study. HH reports grants and non-financial support from Servier, during the conduct of the study. JWV reports Consulting or Advisory role for Ipsen, Novartis, AstraZeneca, Merck, Delcath Systems, Agios, Pfizer, PCI Biotech, Incyte, Keocyt, QED, Pieris Pharmaceuticals, Genoscience Pharma, Mundipharma EDO; Honoraria from Ipsen; and Speakers' Bureau for Novartis, Ipsen, NuCana and Imaging Equipment. MGM has received research grant support from Servier, Ipsen and NuCana. She has received travel and accommodation support from Bayer and Ipsen and speaker honoraria from Pfizer, Ipsen and NuCana. She has served on advisory boards for Celgene, Ipsen, Sirtex and Baxalta.

**Patient consent for publication** Not required.

**Ethics approval** This study has ethical approval from the Greater Manchester Central Research Ethics Committee (reference no. 18/NW/0031) and clinical trial authorisation from the Medicine and Healthcare Products Regulatory Agency.

**Provenance and peer review** Not commissioned; externally peer reviewed.

**ORCID iD**
Zoe Craig http://orcid.org/0000-0001-9930-6648

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
