## [Reviewer comments · BMJ Open]

ARTICLE DETAILS

TITLE (PROVISIONAL)	The NET-02 trial protocol: A multi-centre, randomised, parallel group, open-label, phase II, single-stage selection trial of liposomal irinotecan (nal-IRI) and 5-fluorouracil (5-FU)/folinic acid or docetaxel as second-line therapy in patients with progressive poorly differentiated extra-pulmonary neuroendocrine carcinoma (NEC)
AUTHORS	Craig, Zoe; Swain, Jayne; Batman, Emma; Wadsley, Jonathan; Reed, Nicholas; Faluyi, Olusola; Cave, Judith; Sharma, Rohini; Chau, Ian; Wall, Lucy; Lamarca, Angela; Hubner, R; Mansoor, Wasat; Sarker, Debashis; Meyer, Tim; Cairns, David; Howard, Helen; Valle, Juan; McNamara, Mairéad

VERSION 1 – REVIEW

REVIEWER	Kenta Murotani Kurume University, Japan
REVIEW RETURNED	16-Oct-2019

GENERAL COMMENTS	The manuscript is an important randomized phase 2 selection design trial. Although appropriately described about study planning, treatment rationale and to take forward to future phase 3 trial, there are some concerns remain. - There is concerned about how to calculate the primary endpoint. Calculating the proportion, if censor patients are excluded from the denominator, the estimated PFS may be overestimated. The authors should describe the handling of censor patients when calculating 6M PFS. Because it is also related to the interpretation of Primary endpoint.- Why does the primary endpoint define as a binary variable (progression-free or not at 6month)? It is natural to define the 6M PFS rate by estimating the rate using the Kaplan-Meier method instead of the binary proportion. I wonder why the authors defined the PFS rate as a binary endpoint and thought it was appropriate. Please let me know if you have a reason.
---

REVIEWER	Gerald Prager Medical University Vienna The Reviewer is supported (speakers honorarium or advisory board meetings) Merck Serono, Roche, Amgen, Sanofi, Lilly, Servier, Taiho, Bayer, Halozyme, BMS, Celgene; CECOG
REVIEW RETURNED	13-Nov-2019

GENERAL COMMENTS	The study proposal is well written and addresses an important topic.
--

VERSION 1 – AUTHOR RESPONSE

NET-02 protocol paper response

We would like to thank the reviewers for their positive response and constructive comments. Please find responses to specific comments itemised below.

There is concern about how to calculate the primary endpoint. Calculating the proportion, if censor patients are excluded from the denominator, the estimated PFS may be overestimated. The authors should describe the handling of censor patients when calculating 6M PFS. Because it is also related to the interpretation of Primary endpoint.

Details regarding censoring of patients for the primary endpoint have now been added to the manuscript in the statistical analysis section as follows:

'An individual is defined to have achieved the primary endpoint if they do not progress within the timeframe of treatment start date until 6 months post-randomisation. If an individual dies or is lost to follow-up, without confirmation of disease progression, within 6-months post-randomisation, they will be considered to have not achieved this endpoint and will be censored at the date of death or date last known to be alive and progression-free.'

Why does the primary endpoint define as a binary variable (progression-free or not at 6month)? It is natural to define the 6M PFS rate by estimating the rate using the Kaplan-Meier method instead of the binary proportion. I wonder why the authors defined the PFS rate as a binary endpoint and thought it was appropriate. Please let me know if you have a reason.

The primary endpoint is defined as a binary variable as this is related to the statistical design using an A'Hern design. This means that the sample size was calculated using this binary endpoint approach. Therefore, the analysis will use the binary variable as the primary endpoint to reflect the initial design of the study. Additionally, this binary variable is being used for the primary endpoint because it is simple to understand and monitor. The data monitoring and ethics committee (DMEC), will review the primary endpoint in each DMEC report, therefore using a binary endpoint counting the number of patients achieving and failing to achieve progression-free or not at 6m will allow ease of monitoring because the DMEC will be able to clearly see if we have reached our 'target' PFS rate of 25% with a lower 95% confidence interval >15%. Finally, the progression-free rate at a specified time-point, defined as a binary variable, is a commonly used endpoint in phase II trials. To support this assertion we provide two examples where this endpoint has been utilised in other phase II cancer trials below:

PHILIP, P. A., MAHONEY, M. R., ALLMER, C., THOMAS, J., PITOT, H. C., KIM, G., DONEHOWER, R. C., FITCH, T., PICUS, J. & ERLICHMAN, C. 2005. Phase II study of Erlotinib (OSI-774) in patients with advanced hepatocellular cancer. *J Clin Oncol*, 23, 6657-63.

SCHOFFSKI, P., RAY-COQUARD, I. L., CIOFFI, A., BUI, N. B., BAUER, S., HARTMANN, J. T., KRARUP-HANSEN, A., GRUNWALD, V., SCIOT, R., DUMEZ, H., BLAY, J. Y., LE CESNE, A., WANDERS, J., HAYWARD, C., MARREAUD, S., OUALI, M., HOHENBERGER, P., EUROPEAN ORGANISATION FOR, R., TREATMENT OF CANCER SOFT, T. & BONE SARCOMA, G. 2011. Activity of eribulin mesylate in patients with soft-tissue sarcoma: a phase 2 study in four independent histological subtypes. *Lancet Oncol*, 12, 1045-52.

VERSION 2 – REVIEW

REVIEWER	Kenta Murotani Biostatistics Center, Kurume University
REVIEW RETURNED	28-Dec-2019
GENERAL COMMENTS	The authors have responded appropriate to most of the concerns. There is no comment.